# Life Cycle Environmental Impacts of a Biobased Acrylic Polymer for Leather Production

**DOI:** 10.3390/polym15051318

**Published:** 2023-03-06

**Authors:** Olga Ballús, Meritxell Guix, Grau Baquero, Anna Bacardit

**Affiliations:** 1Cromogenia Units S.A., c/Feixa Llarga 1-3, 08040 Barcelona, Spain; 2Departament d’informàtica i Enginyeria Industrial, Universitat de Lleida, Av. Pla de La Massa 8, 08700 Igualada, Spain

**Keywords:** leather, sustainability, biodegradability, life cycle assessment (LCA), polymers

## Abstract

The aim of this paper was to develop a biopolymer based on raw materials not originating from petroleum chemistry to reduce the environmental impact. To this end, an acrylic-based retanning product was designed where part of the fossil-based raw materials was replaced with biomass-derived polysaccharides. Life cycle assessment (LCA) of the new biopolymer and a standard product was conducted to determine the environmental impact. Biodegradability of both products was determined by BOD_5_/COD ratio measurement. Products were characterized by IR, gel permeation chromatography (GPC), and Carbon-14 content. The new product was experimented as compared to standard fossil-based product, and the main properties of leathers and effluents were assessed. The results showed that the new biopolymer provides the leather with similar organoleptic characteristics, higher biodegradability and better exhaustion. LCA allowed concluding that the new biopolymer reduces the environmental impact of 4 of the 19 impact categories analyzed. A sensitivity analysis was performed where the polysaccharide derivative was replaced with a protein derivative. The analysis concluded that the protein-based biopolymer reduced the environmental impact in 16 of the 19 categories studied. Therefore, the choice of the biopolymer is critical in this type of products, which may or may not reduce the environmental impact.

## 1. Introduction

Leather is a byproduct of the meat industry that is transformed into a valuable product of high durability, stability, comfort, and breathability [1]. It is considered one of the top products of circular economy [2]. Leather also makes less use of materials originating from nonrenewable sources, such as plastic and, therefore, contributes to environmental protection and sustainability [3]. Currently, roughly 10 million metric tons of hides/leather are produced every year [1]; if left untreated, they pose serious problems as a residue. The tanning process also generates byproducts for other industries, such as pharma companies and the food industry [4,5,6]. Leather is, therefore, considered a sustainable product.

The tanning process, however, uses large amounts of chemicals that mostly originate from the petroleum industry. For instance, formaldehyde and phenol for phenol synthetic retanning agents, acrylic acid for acrylic resins, or mineral oil for fatliquoring products, are used. Petroleum is a nonrenewable, finite resource that highly contributes to climate change [7,8]. Some studies estimate its exhaustion by 2050 if the current standard of living is maintained [7]. Moreover, its transformation demands a large amount of energy. Indeed, it presently accounts for 20% of the total energy used in the industry [9] and its treatment involves products and/or residues that harm human health and the environment. In addition, these products depend on the price and availability of petroleum, which are not always consistent [10]. In 2020, petroleum production by the European Union reached its lowest level (18.7 MT), and petroleum import was 96.2%, also in 2020 [11]. That is why petroleum is considered an unsustainable resource [12]. Consequently, petroleum derivatives are considered unsustainable products and should be reduced or avoided where possible. This is why great effort is being devoted to searching for alternative resources [7,13]. On the other hand, biomass or biobased products can be used as substitutes for fossil resources [10]. Currently, biomass is considered the only sustainable source of organic carbon for the industry [13,14].

According to the European Commission Report [15], in the framework of the Lead Market Initiative, biomass-derived products have a high economic and social value on account of several factors, such as reduced dependence on fossil fuels, potential for greenhouse gas emission reduction, potential for sustainable production, potential for better recycling, and better recovery. Moreover, these products are often less toxic, more biodegradable or compostable, consume fewer resources (water, energy), are less hazardous to health, support rural development, and increase industrial competitiveness. The InnProBio factsheet n.1 [16] also indicates some qualities of biomass derivatives. Indeed, a smart use of biomass may lead to improved versions of fossil alternatives or to innovating products, may contribute to lessen environmental impact by reducing greenhouse gases, may reduce product toxicity, residues and dependence on nonrenewable resources such as petroleum, and may promote job creation in the rural environment.

Biomass derivatives may originate from different sources, such as sugars, starch, proteins, natural fibers and wood, lignin derivatives, natural oils and fats, mixed waste, and natural glue [16,17]. Sugars may be incorporated into different polymers to obtain a graft copolymer [18,19,20,21,22]. Currently, proteins can be used as retanning agents either applied alone [23,24,25] or reacted with other polymers [26,27].

In the retanning process, different kinds of products are used that provide different characteristics to the leather. Acrylic resins are widely used at this stage and confer very good properties to the leather, such as high fullness or high tightness. The acrylic resins used in retanning processes are generally polymers of high molecular weight formed by repeating, smaller units called monomers. The properties conferred to the skin mostly depend on the type of monomer used during synthesis and the molecular weight of the resins. Most commonly used monomers are acrylic acid, acrylonitrile, styrene, and maleic acid. These products are usually synthesized by chain growth polymerization, where polymeric growth is due to the reaction of the polymer with a reactive terminal group [28].

Polymerization may involve different types of monomers, with copolymer formation, or the same type of monomer, with homopolymer formation. Copolymers are classified according to the distribution of repeating units [28,29] as: statistical copolymers, random copolymers, alternate copolymers, block copolymers, or graft copolymers. Graft copolymers are part of a defined backbone with randomly distributed branches or side chains different from the main chain. They can be considered branched block copolymers [18,28]. Statistical, random and alternate copolymers usually have intermediate properties as compared to the properties of their monomers, while block or graft copolymers usually have the properties of the monomers used in polymerization and may have unique characteristics due to the bonds formed between their different units. Therefore, these are very interesting products [28,29]. Copolymers can be added with biomass derivatives to obtain graft copolymers, thus improving biodegradability and reducing the environmental impact caused by the synthesis of acrylic resins and derivatives thereof. These biopolymers are abundant, biodegradable and nontoxic, and may react with an initiator to form free radicals and start polymerization [29,30]. 

Extensive bibliography is available on the design of more sustainable products for different industries. They all have one point in common: they replace fossil derivatives with biomass derivatives or natural resources [1,9,13,31,32,33,34,35,36,37,38,39,40]. Adding biomass derivatives to fossil products may improve product biodegradability, has a positive economic impact, and may enhance environmental performance [41]. 

Life cycle assessment (LCA) can be used to determine the environmental impact of a product. LCA is a tool that allows the study and identification of environmental impacts related to the life cycle of a product, organization or process. This is how the product’s life cycle stages that have a greater impact on the environment are determined and able to be improved. LCA allows the redesigning of products or designing new ones, and comparing products [42,43]. LCA is a standard ISO method (ISO 14040: principles and reference framework, and ISO 14044: requirements and directives) that identifies, quantifies and evaluates the environmental impact of a product, process or service at all stages of its life: extraction, production, distribution, end of life. LCA allows the comparison of the environmental impact of a standard acrylic resin to that of a biopolymer, and then establishing the sustainability of the product under study. 

The aim of this paper was to develop a biopolymer based on alternative raw materials not originating from petroleum chemistry in order to reduce the environmental impact. To this end, LCA of the synthesized product was performed and compared with a standard product. The results showed that, in some cases, the new polymer had a greater environmental impact than the standard product. Therefore, adding biomass derivatives to a standard fossil product does not improve all environmental impacts.

## 2. Materials and Methods

### 2.1. Synthesis of BB Graft Copolymer

Acrylic acid as monomer of the polymer was selected to synthesize product BB graft copolymer, while a polysaccharide derivative was selected as a polysaccharide-biomass derivative from agroforestry residues (BPS). The products were synthesized with classic laboratory reagents used for this type of synthesis. 

Synthesis was performed in a four-mouth reactor with a capacity of 1 L. The reactor included an adjustable speed stirrer and a cooling coil.

For synthesis, part of BPS is added with distilled water to the reactor, which is heated to 80 °C with stirring at 100 rpm. When the temperature is reached, the acrylic acid monomer is added for 3 h. The rest of BPS is then added together with the catalyzer for another 3 h, keeping the temperature at 80–85 °C. The reaction goes to completion by adding distilled water together with the catalyzer for full monomer depletion. It is cooled to 50 °C and neutralized with the sodium hydroxide solution.

In order to obtain a satisfactory product of appropriate molecular weight, different syntheses are performed by adding different quantities of BPS and the same quantity of acrylic acid. The reaction is optimized by adding one part of BPS with distilled water at the beginning of the reaction, and then another part of BPS with the acrylic monomer and the catalyzer. BPS is added in varying proportions at each stage.

### 2.2. Product Characterization

#### 2.2.1. Molecular Weight Analysis

Gel permeation chromatography (GPC) was used for molecular weight determination, using the Agilent model 1260 Infinity coupled with 1260 MDS refractive index detector. The products were analyzed using a column set comprising three Ultrahydrogel columns 7.8 × 300 mm (Waters) with 120, 250 and 500 pore sizes. These columns provide the following molecular weight range: 1000–80,000 g/mol, 10,000–400,000 g/mol, and 10,000–1,000,000 g/mol. The columns were packed with hydroxylated polymethacrylate gel, and the particle sizes for Ultrahydrogel 120 and 250 and for Ultrahydrogel 500 were 6 μm and 10 μm, respectively. A 0.1 M sodium nitrate aqueous solution at a flow rate of 0.8 mL/min was used as mobile phase.

#### 2.2.2. IR Analysis

IR spectra in the range of 600–4000 cm^−1^ were measured on a Perkin Elmer Spectrum One. The samples were polymer films onto ZnSe base.

#### 2.2.3. Biobased Carbon Content

The biobased carbon content can be determined by carbon 14 (C-14) analysis. 

The C-14 content of the biopolymer was determined using standard ASTM D6866-18 Method B (AMS) (accelerator mass spectrometry)

C-14 is an unstable isotope produced in the upper layers of the atmosphere when cosmic rays collide with nitrogen. It enters the food chain through photosynthesis (CO_2_), and all living things have a uniform, constant concentration of C-14. C-14 has a half-life of 5730 years and so its concentration is reduced by half every 5730 years. As soon as a living organism dies, it stops taking in new carbon and the C-14 concentration slowly drops until it practically disappears. No C-14 is detected in a fossil carbon sample that is millions of years old. Therefore, knowing the total quantity of total C in the sample and detecting the quantity of C-14 allows determining the C-bio/C-fossil ratio of the sample.

Standard ASTM D6866-18 Method B (AMS) allows measuring C-14 content against the carbon 12 and carbon 13 of the sample, and this is compared with an oxalic acid standard as C-14-free sample.

#### 2.2.4. COD, BOD_5_ and Biodegradability

Chemical oxygen demand (COD) was determined for each of the two products and residual baths thereof after application at the retanning stage. COD analysis was performed with 150-g/L vials heated under reflux for 2 h at 150 °C. Biological oxygen demand (BOD_5_) was also determined for products and residual baths. BOD_5_ was determined by neutralizing the sample with 2% sodium hydroxide up to pH 6–7. Previous COD analysis was performed. A series of dilutions was performed and samples were incubated in darkness at 20 °C for 5 days. The dissolved oxygen concentration was determined before and after incubation. The concentrations of oxygen consumed per liter of water were calculated. Finally, relative biodegradability was calculated with the BOD_5_/COD ratio.

#### 2.2.5. Biodegradability

The biodegradability of a product can be determined by BOD_5_/COD ratio analysis [44,45,46,47]. This ratio does not provide absolute biodegradability results, but rather gives a good insight into product biodegradability, classified as low, medium or high. Biodegradability according to BOD_5_/COD ratio is shown in Table 1 [48].

#### 2.2.6. Life Cycle Assessment (LCA)

Life cycle assessment (LCA), according to ISO 14043 Life cycle impact assessment, was performed to evaluate all environmental impacts associated with the production of the new BB graft copolymer as compared to a standard resin.

The carbon footprint is an impact category of the LCA that provides information about global warming. LCA was performed with OpenLCA 1.8.0 software and the Ecoinvent 3.7.1 database. 

The purpose of life cycle assessment (LCA) was to compare the environmental impacts related to the manufacture of the new biopolymer BB graft copolymer and the standard product.

The functional unit was 1 kg of manufactured product. The reference flow was the kilograms of the two products manufactured on an industrial scale between September 2019 and September 2020. The limits of the system include raw material extraction, raw material transformation, and product production. Distribution, use, and end-of-life were omitted.

Calculations were performed with the impact categories of EF2.0 midppoint for Life Cycle Impact Assessment (LCIA). OpenLCA 1.8.0 software and the Ecoinvent 3.7.1 database were used.

Nineteen impact categories divided into four large groups (resources, health, climate change, and ecosystem) were considered. 

A sensitivity analysis of LCA was also performed with the sustainable acrylic resin to ascertain whether or not the impacts could be reduced. The sensitivity analysis consisted in replacing BPS with another type of protein-based biomass derivative (BPP).

### 2.3. Leather Assessments

Chrome-tanned cattle hide of Spanish origin shaved at 1.4–1.5 mm was used for retanning trials. The hide was cut along the backbone. Standard acrylic and BB were applied on the left and right halves, respectively. A standard application formula shown in Table 2 was used. After retanning, BOD_5_ and COD of the effluents and fastness on dried leathers were analyzed. After dyeing and fatliquoring, physical determinations on dried leathers were assessed.

The following commercial-grade chemicals were used for retanning: anionic dyestuff (color index Acid Brown 83), sulfated neatsfoot oil (75% of active matter), and sulfochlorinated paraffin (67% of active matter). The chemicals used in the operations before and after retanning were classic chemicals customarily used in tanning processes.

Pieces of leather were collected for degree of whiteness and lightfastness tests before the dyeing/fatliquoring stage. Lightfastness and heat resistance tests were performed according to IUF 402 (UNE EN ISO 105-B02) and IUF 412 (UNE EN ISO 17228), respectively, at 102 °C for 100 h. The degree of whiteness was determined with a Datacolor Spectraflash SF-300 colorimeter. Softness according to UNE EN ISO 17235, thickness according to IUP 4 (UNE EN ISO 2589), tensile strength according to IUP 6 (UNE EN ISO 3376), tear load according to IUP 8 (UNE EN ISO 3377-2) with Zwick TMZ2.5/TN1S, and firmness with the Satra PM36 method using a break/pipiness scale, were measured. Organoleptic assessment was performed for color intensity and color levelness.

## 3. Results and Discussion

The acrylic resins used in the tanning process are generally high molecular weight polymers formed by repeating smaller units called monomers. The properties they provide to the leather depend basically on the type of monomers used during the synthesis and the molecular weight of the resins. As mentioned, the most commonly used monomers are acrylic acid, acrylonitrile, styrene, and maleic anhydride. The purpose of this study was to develop an acrylic resin where part of the acrylic acid monomer was replaced with a biomass-derived polysaccharide (BPS).

A radical polymerization reaction between an acrylic acid monomer and BPS was carried out. A graft copolymer with a defined backbone and randomly distributed branches or side chains different from the main chain was obtained. The possible synthesis reaction of the new BB graft copolymer is shown in Figure 1(1) and the acrylic acid radical polymerization is shown in Figure 1(2) [29,41] In order to better understand the radical polymerization, acrylic acid is marked in red.

### 3.1. Product Characterization

GPC determination was performed according to the previous section. The number of the average molecular weight (M_n_) is the statistical average molecular weight of all the polymer chains in the sample. The weight average of the molecular weight (M_w_) was obtained taking into account the contributions of each molecular weight. Finally, the polydispersity index (PDI) was defined by Equation (1). The results are shown in Table 3.
(1)$$\mathrm{PDI}=\frac{M_{w}}{M_{n}}$$

Standard acrylic had a higher molecular weight (M_w_) but a lower M_n_ value. Because its PDI value was higher than that of BB graft copolymer, the latter was less polydisperse. This difference in PDI values may be due to the fact that a different catalyst has been used for the synthesis of BB graft copolymer, which is more selective and allows narrowing the dispersion of the molecular weights, providing a lower PDI value.

IR spectra data allowed the identification of the functional groups that characterize polymers and their constituent monomers. IR spectra of sustainable acrylic resin BB and of acrylic acid are shown in Figure 2 and Figure 3, respectively. The main absorption bands of the functional groups present in acrylic acid were identified. The band of the –OH bond of the carboxyl group appeared at 3063 cm^−1^, followed by the CH_2_ band at 2661 cm^−1^. The C=O group band appeared at 1704 cm^−1^, the C–H band appeared at 1434 cm^−1^, and the doublet appeared at 1298 cm^−1^ and 1243 cm^−1^ due to C–O bond stretching from esters [49].

The spectrum of the product obtained was compared to that of acrylic acid. The spectrum of the polymer synthesized in the laboratory showed a wide, intense absorption band at 3376 cm^−1^, related to –OH bond stretch vibration. The CH_2_ band and the carbonyl group band appeared at 2934 cm^−1^ and 1707 cm^−1^, respectively. Finally, the absorption band representing C–O–C interaction appeared at 1152 cm^−1^, thus suggesting the formation of the ester bond not present in acrylic acid [49].

The results of biobased carbon content, COD, BOD_5_ and sample biodegradability are shown in Table 4.

Biobased carbon content is a measure of the amount of biomass-derived carbon (C-14) in a product as compared to its total organic carbon content, and is expressed in percentage. Petroleum-derived products do not contain carbon-14. Standard acrylic has 0% biobased carbon content and, therefore, is not a biobased product (it is made 100% from petrochemical resources). BB has 46% biobased carbon content, meaning that 46% of the product’s carbon originates from biomass resources and 54% from petroleum-derived materials. 

Unlike the BOD_5_ values obtained, the COD values for both products were similar (9% variation). Both COD and BOD are parameters that allow determining the oxygen demand strength of sewage, COD being a chemical oxidation process and BOD being a biological oxidation process. BOD_5_ values are lower than COD values because COD measures the oxygen demand for the decomposition of both organic and inorganic materials in sewage. Higher BOD_5_ values suggest that more oxygen is being consumed by the sample.

Finally, the BOD_5_/COD ratio allows obtaining an estimated value of the sample’s biodegradability. The value for standard acrylic was 0.01 (non-biodegradable), while the value for BB was 0.71 (biodegradable). These results evidenced that adding the biomass-derived polysaccharides to an acrylic resin substantially improves biodegradability.

### 3.2. Leather Assessments

Products were applied as sole products at the retanning stage according to a standard formulation in Table 2.

BOD_5_ and COD values in residual baths were analyzed to determine bath biodegradability. The results are shown in Table 5.

The residual baths of standard acrylic showed a lower BOD_5_ value as compared to product BB, a trend similar to that obtained with the BOD_5_ of the product. Regarding biodegradability, biodegradable baths with BB (BOD_5_/COD = 0.8) and non-biodegradable baths with standard acrylic (BOD_5_/COD = 0.1) were obtained.

Both the degree of whiteness and lightfastness (or accelerated aging) were determined to prevent yellowing issues, mainly on light or colorless leather articles. The parameters indicated for degree of whiteness are: L* (luminosity, where the greater the value, the greater the luminosity), a* (red/green, where +a* tends to red and −a* tends to green), and b* (yellow/blue, where +b* is yellower and −b* is bluer). The values of the lightfastness test are expressed in blue scale. An 8-point rating scale was used, where 0 is extremely poor lightfastness and 8 is excellent lightfastness. Heat resistance values were expressed in a 5-point grey scale, where 1 is the minimum value and 5 the maximum value. The standard acrylic resin was slightly whiter with higher L* = 82.91 vs. 80.81 value for BB, and similar a* = −3.7 for the standard acrylic vs. −3.87 for BB. b* values were also similar with b* = −2.14 for the standard acrylic and −1.87 for BB. On the other hand, very good lightfastness and heat resistance values were obtained. Both products were within specifications (≥3) and little oxidizable by environmental factors such as light and/or temperature. Therefore, introducing the biomass derivative does not decrease the good fastness properties of the standard acrylic resin. Fastnesses results are shown in Table 6.

The degrees of softness, thickness, firmness, physical resistances and color properties in crust hides are shown in Table 7. The percent variation between samples for softness, thickness and resistances was also calculated. On the other hand, tightness was expressed according to the Satra break/pipiness scale, where 1 is the maximum value and 8 is the minimum value, and color levelness was expressed from 1 to 5, where 1 is the minimum value and 5 is the maximum value.

BB was softer, although slightly less full, with improved elongation and tear load as compared to standard acrylic. The tensile strength of the new biopolymer was slightly lower than that of the standard product. While good firmness was provided by standard acrylic, BB was one point better and reached the maximum value. On the other hand, BB provided high color intensity—the L* value was lower (less luminosity) without decreasing color levelness. 

### 3.3. Life Cycle Assessment

According to Section 2.2.6, the results are shown in Figure 4 as a stacked bar chart with 50% contribution as red line.

Biopolymer BB has a greater environmental impact in the categories related to (non-fossil) natural resources, toxicity parameters, and human health. This is due to sugarcane cultivation, the origin of the polysaccharide derivative. Sugarcane cultivation has a high environmental impact due to the high use of fertilizers, herbicides or pesticides, which increase eutrophication, acidification, or waters with low concentrations of dissolved oxygen [50,51,52,53,54]. Moreover, because sugarcane is burnt to facilitate harvesting, it releases greenhouse gases that impair human health and pollute the air [54]. In turn, the standard resin has a greater impact in all the categories related to fuels and fossil resources, which are obtained from petroleum. In general, a greater impact of BB in 14 of the 19 categories selected was observed. In the non-carcinogenic effect category, BB was much better than the standard acrylic resin because sugarcane cultivation uptakes Zn, Cu and Ni. BB was also better than the standard resin in the climate change and carbon footprint categories, reduced by 9%.

On account of the high environmental impact of the polysaccharide derivative, a sensitivity analysis was performed where the polysaccharide derivative was replaced with a protein derivative (AP). The results are shown in Figure 5 as a stacked bar chart with 50% contribution as red line. 

In this case, there is an improvement in 17 of the 19 categories studied and, therefore, a clearly reduced environmental impact of AP vs. standard is obtained. The incorporation of protein derivatives improved the results vs. polysaccharide derivatives, where only 5 of the 19 categories improved. Product AP decreased the carbon footprint by 47%, as compared to 9% by product BB. This is because the protein derivative used is a byproduct, which causes a much smaller environmental impact.

## 4. Conclusions

The new biopolymer had properties similar to those of a standard resin in terms of fullness, tightness, physical resistances and fastnesses, with a few slight differences: BB provides slightly less fullness since thickness decreases by 7%, and more softness with 6% increase compared with a standard product. Regarding physical resistance, the new biopolymer showed higher tear load (+14%)and higher elongation (+17%) than the standard product. BB showed slightly lower tensile strength (−8%) than the standard product. Regarding grain firmness, the new product provided better results than the standard product, reaching the maximum value (1).The new resin showed greater biodegradability of both product and sewage, with the ratio BOD_5_/COD of the product 0.71 vs. 0.01, the ratio for the standard product. The new product showed an improved fixation and, therefore, less COD. Life cycle assessment (LCA) allowed measurement of the environmental impact of this new product and comparison to a standard resin. LCA allowed the conclusion that the use of biopolymers reduces the environmental impact in terms of carbon footprint by 9%, but also showed that the standard acrylic resin had a lower environmental impact in 14 of the 19 impact categories studied. A sensitivity analysis was also performed where the polysaccharide derivative was replaced with a protein-derived biopolymer. In this case, a reduced environmental impact was obtained in all but two LCA categories. Therefore, the biopolymer must be carefully selected bearing in mind the environmental impact likely to be caused by this product, which, in some cases, may be higher than the environmental impact caused by acrylic acid.

## Figures and Tables

**Figure 1 polymers-15-01318-f001:**
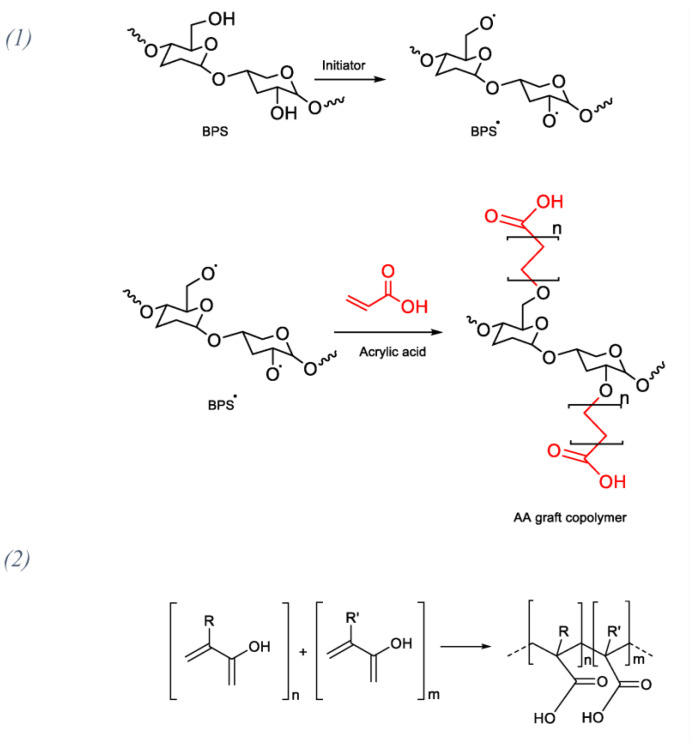
(**1**) Possible synthesis of the new BB graft copolymer; (**2**) Acrylic acid radical polymerization.

**Figure 2 polymers-15-01318-f002:**
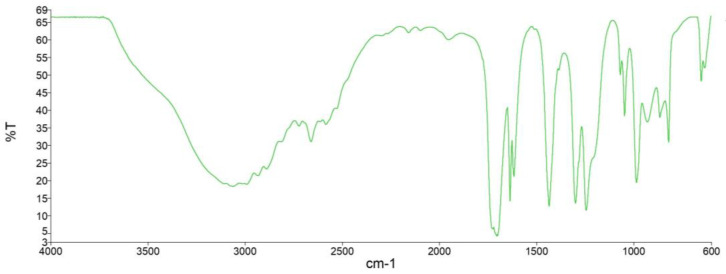
IR spectrum of BB graft copolymer.

**Figure 3 polymers-15-01318-f003:**
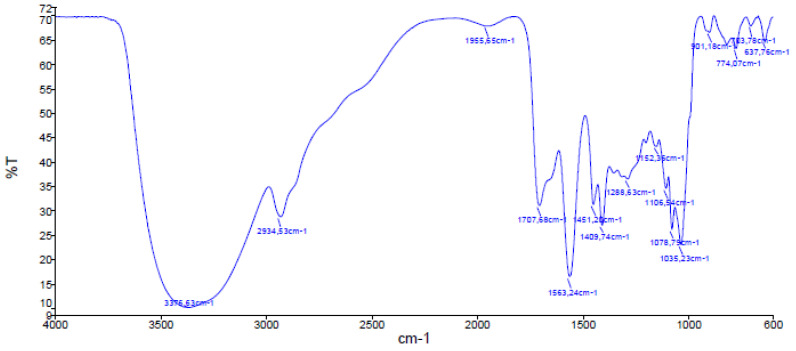
IR spectrum of acrylic acid.

**Figure 4 polymers-15-01318-f004:**
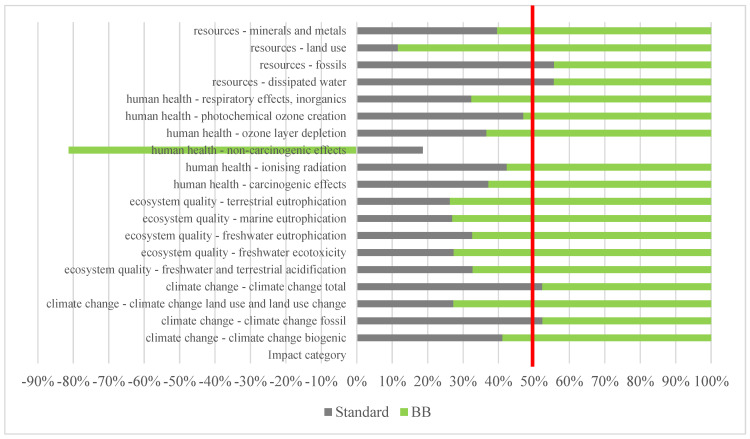
Contribution of standard product and BB product to LCA’s impact categories.

**Figure 5 polymers-15-01318-f005:**
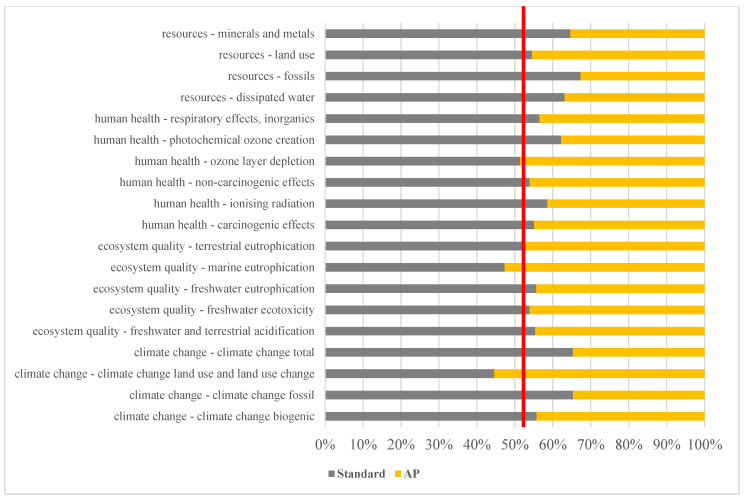
Contribution of standard product and AP product to LCA’s impact categories.

**Table 1 polymers-15-01318-t001:** Biodegradability according to BOD_5_/COD ratio.

BOD_5_/COD	B < 0.2	0.2 > B < 0.45	B > 0.45
Biodegradability	Low biodegradability	Medium biodegradability	High biodegradability

**Table 2 polymers-15-01318-t002:** Standard application formula.

Process	Chemicals	%	Time (min)	T (°C)	Remarks
Washing	Water	100		35	
	Non-ionic surfactant	0.2			
	Formic acid	0.2	30		pH = 3.5
Drain					
Neutralization	Water			35	
	Sodium formate	2	15		
	Sodium bicarbonate	0.5	60		pH = 5.5
Drain and wash			10		
Retanning	Water	100		40	
	5% product (related to active matter)		60		
Assessments Effluents: COD, BOD_5_ Leathers: Fastness tests and whiteness degree on dried leathers					
Dyeing	Water	100		45	
	Dyestuff (Acid Brown 83)	2	45		
Fatliquoring	Sulfochlorinated oil	5			
	Sulfated oil	5	60		
	Formic acid	2	60		pH = 3.8
Drain, wash					
Mechanical operations	Sammy-set out, vacuum 2′ at 50 °C, stake				
Assessments Physical determinations					

**Table 3 polymers-15-01318-t003:** Number average molecular weight, weight average molecular weight, and polydispersity index of the standard acrylic polymer and BBgraft polymer with BPS.

Sample	M_n_ (g/mol)	M_w_ (g/mol)	PDI
Standard acrylic	65,910	734,579	11.15
BBBB graft copolymer	192,332	582,634	3.03

**Table 4 polymers-15-01318-t004:** Biobased carbon content, COD, BOD_5_ and biodegradability of the products studied.

Sample	Biobased Carbon Content (%)	BOD_5_ (mg/L)	COD (mg/L)	BOD_5_/COD
Standard acrylic	0	2867	304.000	0.01
BB graft copolymer	46	195.250	276.000	0.71

**Table 5 polymers-15-01318-t005:** COD, BOD_5_ and biodegradability in residual baths.

Sample	COD (mg/L)	BOD_5_ (mg/L)	BOD_5_/COD
Standard acrylic	4330	325	0.1
BB	3340	2560	0.8

**Table 6 polymers-15-01318-t006:** Degree of whiteness and fastness tests.

Sample	Degree of Whiteness			Lightfastness **	Heat Resistance (GS)
	L*	a*	b*		
Standard acrylic	82.91	−3.71	−2.14	4Y	3/4
BB	80.81	−3.87	−1.87	4Y	3/4

** Y: Yellowing.

**Table 7 polymers-15-01318-t007:** Degrees of softness, thickness, firmness, physical resistance, color intensity and color levelness in crust hides.

Sample	Softness	Thickness (mm)	Tensile (MPa)	Elongation (%)	Tear (N)	Firmness	Color Intensity (L*)	Color Levelness
Standard acrylic	3.5	1.5	26.85	17.68	159.4	2	55.82	5
BB	3.7	1.4	24.65	20.67	182.1	1	52.49	5
Variation	+6%	−7%	−8%	+17%	+14%	-	-	-

## Data Availability

Data available on request from the authors.

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
