# Peer review of "Life Cycle Environmental Impacts of a Biobased Acrylic Polymer for Leather Production"

_polymers, 2023, doi:10.3390/polym15051318_

Round 1
Reviewer 1 Report
In my opinion the paper with title: “
Life cycle environmental impacts of a biobased acrylic polymer 2 for leather production” could be published in the journal Polymer if the authors will make the corrections and specifications suggested:
1. Please specify in the article how Biobased acrylic polymer is obtained (what are the raw materials)
2. Please specify if equation 1 defines PDI or PD and make the correction in the article.
3. Please establish a symbol for biodegradability, possibly B.
In Table 1 Biodegradability cannot be expressed as shown in Table 1. please correct with:
0.2< B < 0.7 respectively 0.7 <B< 0.8
3. Please specify according to which standard or evaluation you made the classification from Table 1. I did not find the reference [48] that specifies this classification. Please enter a more conclusive reference.
4. In chapter 2.2.6. Life cycle assessment (LCA) the assessment of the environmental impact of the Life Cycle Analysis, LCA, has the standardized name, according to ISO 14043 -Life cycle impact assessment. Please insert in the article this name.
5. Please define in Chapter 2.2.6., as clearly as possible, what are the premises of the life cycle analysis (purpose and objectives), what is the process tree from where it starts (raw materials, unit processes, what is the product, where makes distribution and storage - capitalization).
6. For a better understanding of what you represented graphically (Graph 1 and 2),
please also specify which are the consumption of materials, the categories of emissions and energy consumption that you have identified throughout the study, and which are the categories of impact that you have identified.
7. I recommend that you present a table with the inventory data and a table with the weighted data, so that it is easier to follow.
8. Please rename Graph 1 and Graph 2, etc. with Figure 1 and Figure 2.
Author Response
Response to Reviewer 1
Life cycle environmental impacts of a biobased acrylic polymer 2 for leather production” could be published in the journal Polymer if the authors will make the corrections and specifications suggested:
- Please specify in the article how Biobased acrylic polymer is obtained (what are the raw materials)
- Please specify if equation 1 defines PDI or PD and make the correction in the article.
Equation 1 defines PDI, changed.
- Please establish a symbol for biodegradability, possibly B.
In Table 1 Biodegradability cannot be expressed as shown in Table 1. please correct with:
0.2< B < 0.7 respectively 0.7 <B< 0.8
Corrected.
- Please specify according to which standard or evaluation you made the classification from Table 1. I did not find the reference [48] that specifies this classification. Please enter a more conclusive reference.
The reference has been changed.
- In chapter 2.2.6. Life cycle assessment (LCA) the assessment of the environmental impact of the Life Cycle Analysis, LCA, has the standardized name, according to ISO 14043-Life cycle impact assessment. Please insert in the article this name.
Done.
- Please define in Chapter 2.2.6., as clearly as possible, what are the premises of the life cycle analysis (purpose and objectives), what is the process tree from where it starts (raw materials, unit processes, what is the product, where makes distribution and storage - capitalization).
Done.
- For a better understanding of what you represented graphically (Graph 1 and 2),
please also specify which are the consumption of materials, the categories of emissions and energy consumption that you have identified throughout the study, and which are the categories of impact that you have identified.
- I recommend that you present a table with the inventory data and a table with the weighted data, so that it is easier to follow.
- Please rename Graph 1 and Graph 2, etc. with Figure 1 and Figure 2.
Graph 1 was changed by Figure 4 and Graph 2 by Figure 5.
As for the question 1, 7, and 8 we cannot provide more information due to the fact that ther is a confidentiality agreement between the university and the company in order to protect the invention for a new business of the company.
Thank you very much for your kindly revision.

Reviewer 2 Report
1) For “AA” in the manuscript, it appears in “AA graft copolymer”, “AA grafted polymer”, “copolymer graft AA”, “AA copolymer graft”, or “AA graft polymer”. Please be consistent. Also, AA is usually the abbreviation for acrylic acid, the monomer, so it could be confusing for readers. It is also confusing when “acrylic acid” is used as a synthesized polymer but also as a monomer. Words like “prepolymer” might be considered.
2) In Table 3, it was shown that the standard acrylic shows a higher PDI than AA. The authors might want to explain more in details how grafting affect the polymer properties. Is there crosslinking in the polymers?
3) In Figure 1, the authors put the free radicals on oxygen as the reaction position, during “the free radical polymerization reaction between an acrylic acid monomer and BPS”. The figure is misleading because it portrayed the grafting reaction instead of polymerization (starting from monomer). In addition, the radical should be on carbon for the polymerization, and chain transfer reaction of acrylic acid could lead to branches, instead of grafting. Furthermore, the reaction between BPS (polysaccharide) and polyacrylic acid should be esterification reactions (nucleophilic substitution) instead of free radical reactions. Please double check the reaction mechanism (citing some references) and redraw Figure 1 for accuracy.
4) In line 245, Figure 2 and Figure 3 are for IR spectra, instead of 1 and 2.
5) For IR analyses, authors need to stress that the doublet peaks are from esters in line 249.
6) Section 2.2.1 should not be highlighted in bold, while “Leather assessments” in line 280 needs to be.
7) How is the addition of BPS affecting the cost/price of the resin?
8) What are the main advantages of AA compared with other biodegradable strategies? Is there any uniqueness of the polysaccharide in the research compared to other biomass derivatives used in graft copolymers?
Author Response
Response to Reviewer 2
1) For “AA” in the manuscript, it appears in “AA graft copolymer”, “AA grafted polymer”, “copolymer graft AA”, “AA copolymer graft”, or “AA graft polymer”. Please be consistent. Also, AA is usually the abbreviation for acrylic acid, the monomer, so it could be confusing for readers. It is also confusing when “acrylic acid” is used as a synthesized polymer but also as a monomer. Words like “prepolymer” might be considered.
The new product is not a prepolymer. Is a copolymer. The new polymer is called BB graft copolymer. All the text has been changed with the new name.
2) In Table 3, it was shown that the standard acrylic shows a higher PDI than AA. The authors might want to explain more in details how grafting affect the polymer properties. Is there crosslinking in the polymers?
This difference in PDI values may be due to the fact that a different catalyst has been used for the synthesis of BB graft copolymer, which is more selective and allows narrowing the dispersion of the molecular weights, providing a lower PDI value. The paragraph has been added in Line 306.
3) In Figure 1, the authors put the free radicals on oxygen as the reaction position, during “the free radical polymerization reaction between an acrylic acid monomer and BPS”. The figure is misleading because it portrayed the grafting reaction instead of polymerization (starting from monomer). In addition, the radical should be on carbon for the polymerization, and chain transfer reaction of acrylic acid could lead to branches, instead of grafting. Furthermore, the reaction between BPS (polysaccharide) and polyacrylic acid should be esterification reactions (nucleophilic substitution) instead of free radical reactions. Please double check the reaction mechanism (citing some references) and redraw Figure 1 for accuracy.
A new reaction mechanism has been added in page 7.
4) In line 245, Figure 2 and Figure 3 are for IR spectra, instead of 1 and 2.
Changed. Line 258.
5) For IR analyses, authors need to stress that the doublet peaks are from esters in line 249.
Done.
6) Section 2.2.1 should not be highlighted in bold, while “Leather assessments” in line 280 needs to be.
Done.
7) How is the addition of BPS affecting the cost/price of the resin?
There is no difference in the cost of the new resin. The company was aiming to obtain a more biodegradable resin without affecting the cost to maintain the same level of sales as using the conventional one.
8) What are the main advantages of AA compared with other biodegradable strategies? Is there any uniqueness of the polysaccharide in the research compared to other biomass derivatives used in graft copolymers?
When we started the research, we tried to add proteins and polysaccharides. When using proteins, we obtained a resin harder. When using polysaccharides, the obtained resin was softer, more suitable to apply in the leather sector.
Thank you very much for your kindly revision.

Reviewer 3 Report
Manuscript ID: polymers-2137515
Title: Life cycle environmental impacts of a bio-based acrylic polymer for leather production
Journal: Polymers
Comments to authors:
The aim of this paper was to develop a biopolymer based on raw materials not originating from petroleum chemistry to reduce the environmental impact. To this end, an acrylic-based retanning product was designed where part of the fossil-based raw materials was replaced with biomass-derived polysaccharides. Life cycle assessment (LCA) of the new biopolymer and a standard product was conducted to determine the environmental impact. Biodegradability of both products was determined by BOD5/COD ratio measurement. IR, gel permeation chromatography (GPC), and Carbon-14 content characterized products. The new product was experimented as compared to standard fossil-based product, and the main properties of leathers and effluents were assessed. Although extensive work has been performed but the novelty can be questioned. However, before it can be considered for publication, please address the following comments for a revision:
1) The Abstract should be enriched with the brief details of the experimental methodology. The problem to be addressed in this study should also be highlighted in the Abstract.
2) Please highlight the novelty in the Abstract also.
3) The authors should also present some quantitative results in the Abstract.
4) English proofreading is required for grammatical mistakes and typos.
5) The novelty and significance of the present work should be highlighted in the last paragraph of the Introduction section.
6) The authors are recommended to add latest relevant literature review on such works.
7) What is the need for this work? Is this work helpful for practical applications? Which applications?
8) The literature review should be improved by adding latest references and discussion.
9) Work methodologies need more discussion.
10) Country name of MDS refractive index detector?
11) Results section should be defended using technical reasons and relevant references.
12) Overlapping of text in Figure 3 should be avoided?
13) Instrumental analysis of the fabricated samples is also required.
14) More technical discussion to the presented experimental results should be added.
15) There are no critical review/discussions before the Conclusions. Authors should add it.
16) Conclusions should be refined and briefly presented. More numerical results should be added.
17) What are the limitations of the present study? Please mention them in the manuscript.
18) The authors can add the future recommendations based on the present study.
Author Response
Response to Reviewer 3
Comments to authors:
The aim of this paper was to develop a biopolymer based on raw materials not originating from petroleum chemistry to reduce the environmental impact. To this end, an acrylic-based retanning product was designed where part of the fossil-based raw materials was replaced with biomass-derived polysaccharides. Life cycle assessment (LCA) of the new biopolymer and a standard product was conducted to determine the environmental impact. Biodegradability of both products was determined by BOD5/COD ratio measurement. IR, gel permeation chromatography (GPC), and Carbon-14 content characterized products. The new product was experimented as compared to standard fossil-based product, and the main properties of leathers and effluents were assessed. Although extensive work has been performed but the novelty can be questioned. However, before it can be considered for publication, please address the following comments for a revision:
- The Abstract should be enriched with the brief details of the experimental methodology. The problem to be addressed in this study should also be highlighted in the Abstract.
- Please highlight the novelty in the Abstract also.
- The authors should also present some quantitative results in the Abstract.
- English proofreading is required for grammatical mistakes and typos.
- The novelty and significance of the present work should be highlighted in the last paragraph of the Introduction section.
- The authors are recommended to add latest relevant literature review on such works.
- What is the need for this work? Is this work helpful for practical applications? Which applications?
- The literature review should be improved by adding latest references and discussion.
- Work methodologies need more discussion.
- Country name of MDS refractive index detector?
- Results section should be defended using technical reasons and relevant references.
- Overlapping of text in Figure 3 should be avoided?
- Instrumental analysis of the fabricated samples is also required.
- More technical discussion to the presented experimental results should be added.
- There are no critical review/discussions before the Conclusions. Authors should add it.
- Conclusions should be refined and briefly presented. More numerical results should be added.
- What are the limitations of the present study? Please mention them in the manuscript.
- The authors can add the future recommendations based on the present study.
We did changes in overall manuscript.
Thank you very much for your kindly revision.

Round 2
Reviewer 1 Report
In my opinion the paper with title: "Life cycle environmental impacts of a biobased acrylic polymer for leather production" could be published in the journal Polymers. The authors made the corrections in accordance with the requirements.
Author Response
Thank you for your kindly review.
As far as I can see in the review report form, no more changes are needed.